# Measurements of metastable helium in Earth's atmosphere by resonance lidar

Bernd Kaifler [1] ✉, Christopher Geach[2], Hans Christian Büdenbender [1], Andreas Mezger[1] & Markus Rapp [1]

Monitoring and predicting space weather activity is increasingly important given society's growing reliance on space-based infrastructure but is hampered by a lack of observational data. Airglow at 1083 nm from metastable helium He(2³S) in the thermosphere has long been a target for remote-sensing instruments seeking to fill that gap; however, passive measurements of He(2³S) fluorescence are limited by low brightness, and interpretation of these observations is complicated by the > 500 km depth of the He(2³S) layer. Here, we demonstrate a lidar instrument that is able to stimulate and detect He(2³S) fluorescence, and we present measured profiles of He(2³S) density. These measurements provide crucial validation to space weather models, support predictions of peak number density ( ∼ 1 cm⁻³) and the dependence of density on altitude, solar zenith angle, and season, and extend by a factor of 4 the maximum probed altitude range by an atmospheric profiling lidar. These measurements open the door for the development of more sophisticated lidars: by applying well-established spectroscopic lidar techniques, one can measure the Doppler shift and broadening of the He(2³S) line, thereby retrieving profiles of neutral wind speed and temperature, opening a window for studying space weather phenomena.

Due to an ongoing lack of observational capabilities, our ability to model and make predictions about the upper atmosphere of the Earth lags far behind that in the lower atmosphere[1–3]. This is despite the substantial impact of space weather phenomena on the performance and reliability of space-borne and ground-based technological systems. For example, space weather events can cause significant interference in modern navigation and communication systems, leading to increased safety risks and economic loss, and endangering human life[4–6]. Space weather in the thermosphere is especially challenging to predict, given its close coupling to ionospheric and magnetospheric dynamics and its sensitivity to solar forcing from above and wave activity from below[7]; at the same time, the thermosphere is of particular importance, given its impact on satellite orbits and performance. There is, therefore, a pressing need to improve our measurement capabilities in the thermosphere, particularly measurements of wind, temperature, and neutral density, to improve our understanding of

space weather phenomena, to serve as empirical validation for space weather models, and to provide real-time data assimilation to improve predictions. This is of particular importance as we approach the next solar maximum, predicted for the 2024–2025 period[8,9].

An enticing target for such observations is metastable helium, He(2³S), which is produced through the impact of photoelectrons with >19.8 eV on ground state He, though recombination of He⁺ in the topside ionosphere may also play a significant role[10]. Penning ionization effectively quenches He(2³S) below 250 km, while photoionization is the dominant destruction mechanism at high altitudes[11]; the result is a He(2³S) layer that extends from ∼250 to 800 km or above. The transition to the ground state through single-photon processes is forbidden, leading to a long natural lifetime of ∼7800 s. Further excitation to the 2³P state gives rise to fluorescence, producing a detectable airglow signal when sufficiently illuminated. As shown in Fig. 1, He(2³P) is a triplet state, but the 1083.025 nm and 1083.034 nm lines blend

¹Deutsches Zentrum für Luft- und Raumfahrt, Institut für Physik der Atmosphäre, Oberpfaffenhofen, Germany. ²Deutsches Zentrum für Luft- und Raumfahrt, Institut für Solar-Terrestrische Physik, Neustrelitz, Germany. ✉e-mail: Bernd.Kaifler@dlr.de

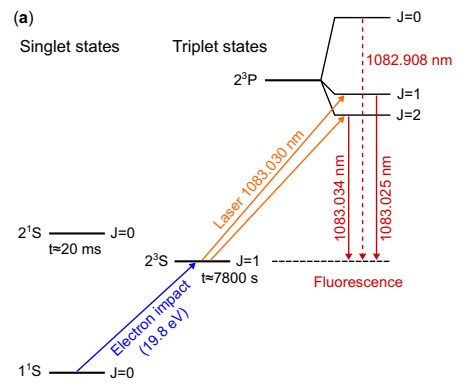

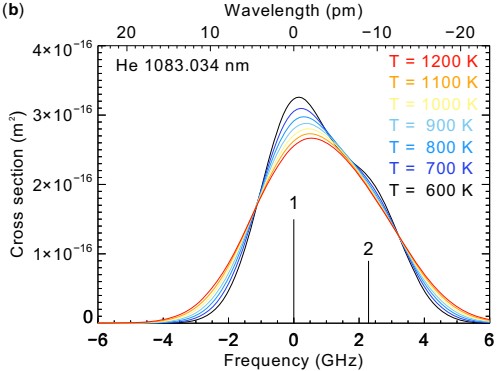

**Fig. 1 | He(2³S) energy levels and fluorescence spectrum. a** Energy levels and transitions in He that are relevant to the He lidar. **b** Theoretical Doppler-broadened fluorescence spectrum of He(2³S) at 1083 nm wavelength for a range of temperatures. The two transitions $2^3P_2 \rightarrow 2^3S_1$ (1) and $2^3P_1 \rightarrow 2^3S_1$ (2) blend together while the third transition $2^3P_0 \rightarrow 2^3S_1$ (not shown) is separated by 126 pm (29.6 GHz).

together at thermospheric temperatures; this combined line is the target of this paper.

Naturally occurring He(2³S) airglow, excited by solar illumination, was first observed more than 60 years ago during a particularly strong aurora over Moscow[12]. Observations with progressively higher spectral resolving power were made over the following decades, but the interpretation of the observations was hampered by the neighboring Q-branch of the OH airglow[13–15]. In 1995–1996, the first measurements capable of independently resolving He(2³S) and the OH line were made[16,17]. Models of thermospheric dynamics have similarly improved, and He(2³S) airglow measurements have offered important insights, for example, into the magnitude of the photoelectron and solar EUV flux, among others[10,11,18]. But observations of He(2³S) fluorescence have lagged behind model development and are now incapable of providing validation. Furthermore, passive measurements of He(2³S) airglow are fundamentally limited in their utility. They represent a vertical integration through the full He(2³S) layer and, as such, imply an average of hundreds of km vertically. Researchers have investigated the airglow brightness or spectral shape as a function of solar zenith angle (SZA) in order to infer information on the vertical distribution and temperature of He(2³S)[13,16,17] since only illuminated He(2³S) contributes to the observed signal, but such attempts unavoidably confound temporal changes in He(2³S) parameters with vertical variations.

Two decades ago, an alternative approach was proposed[19]: to probe the He(2³S) layer with a powerful pulsed laser, stimulating artificial fluorescence and making time-resolved measurements of the backscatter that can be converted into height-resolved measurements of He(2³S) density. Such resonance lidars are a commonly used technique for probing the mesosphere/lower thermosphere region and retrieving profiles of wind and temperature with high resolution and accuracy[20,21] but have so far been limited to metallic layers at altitudes up to 180 km[22]. Extending the technique to He(2³S) would increase the altitude range by a factor of 4 or more, but attempts to do so have been thus far unsuccessful[23].

Here we describe an instrument with which lidar measurements of He(2³S) fluorescence have been achieved and the key advancements that made this detection possible; we present the initial measurements from this instrument, including measured profiles of He(2³S) density; and we discuss the potential improvements and future capabilities of such an instrument for space weather monitoring and observations.

## Results

A total of 159 hours of data were collected during 11 nights between January 13 and March 4, 2022 at the German Aerospace Center in Oberpfaffenhofen, Germany (48.1°N, 11.3°E). These data were sometimes contaminated by the presence of low-level fog and/or clouds; omitting these periods and periods with SZA less than 97°, 117 h of data remain.

The data are sensitive to the presence of He(2³S) in two ways: an 'active' measurement, comprising the laser-synchronous return signal, and a 'passive' measurement, comprising the range-independent background measured by the lidar.

### Active measurement of He(2³S) density profiles

For the active measurement, we calibrate the system efficiency based on Rayleigh return from the lower atmosphere, and convert measured backscatter counts in 100 km vertical bins to He(2³S) density profiles, following the lidar equation (see Methods section), assuming a scattering cross-section for He(2³S) corresponding to a temperature of 1000 K. Uncertainty in this conversion process contributes 30% relative uncertainty to the calculated He(2³S) densities. Given an expected seasonal variation in He(2³S) densities[10,24] and a previously observed morning/evening asymmetry[14], we separate the measurements into two periods, January 13–February 14, 2022 and February 23–March 4, 2022, and split the measurements within each period into evening and morning, with solar midnight as the dividing point. Since measurements were made over a span of months, the position of the sun was a more important driver of variability than either local or universal time; the data are therefore binned/plotted as a function of SZA rather than time. Data with SZA less than 97° exhibit significantly higher background noise due to Rayleigh scattering of sunlight in the lower atmosphere and are therefore excluded. We divide the remaining data into four bins to evaluate the dependence of He(2³S) density on SZA: 97–110°, 110–120°, 120–130°, and >130°. The resulting He(2³S) density profiles are shown in Fig. 2.

### Passive measurement of He(2³S) airglow

The passive measurement consists of the return signal from altitudes between 100 and 250 km. At these heights, contributions from Rayleigh scattering are negligible, and no metastable helium is expected to be present[10], so we take the return signal over these altitudes as a proxy for the range-independent background. In the middle of the night, the contribution to the background from the sky drops to almost zero, and this passive measurement is dominated by detector noise (~70 Hz). At smaller SZA the metastable helium layer is illuminated by the sun, and natural fluorescence occurs, producing a substantial increase in the observed background (see Fig. 3). This is comparable to the previous He(2³S) measurements described in the introduction, though the narrow bandpass of the receiver optics provides stronger out-of-band rejection and higher signal-to-noise than previous instruments.

## Discussion

The morning He(2³S) density profiles in Fig. 2 show a distinct detection of He(2³S) and exhibit several characteristics that validate model predictions[10]. The peak amplitude is within the predicted range; the

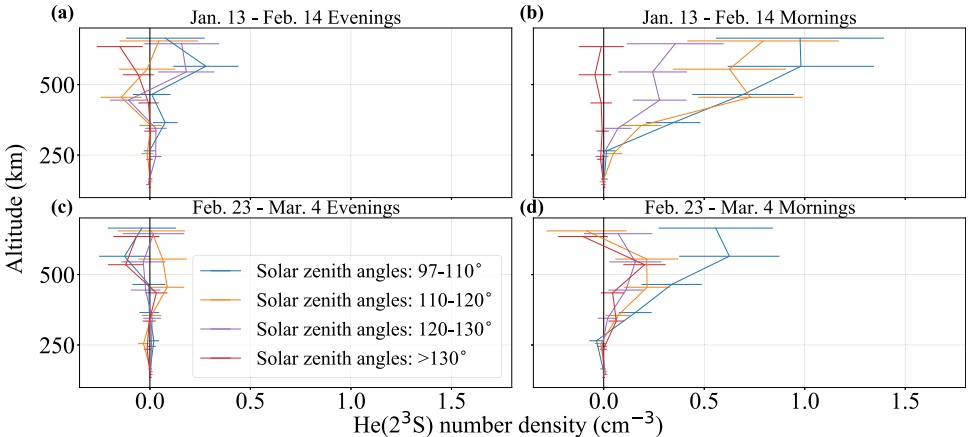

**Fig. 2 | Active He(2³S) measurements.** Evening (**a**, **c**) and morning (**b**, **d**) He(2³S) density profiles measured over six nights between January 13 and February 14, 2022 (**a**, **b**) and five nights between February 23 and March 4, 2022 (**c**, **d**). The data are binned and plotted according to solar zenith angle (see legend). Note that the data points represent vertical bins of 100 km and are plotted with ±5–15 km vertical offsets to avoid overlap. The error bars represent 1σ uncertainties, as described in the Methods section.

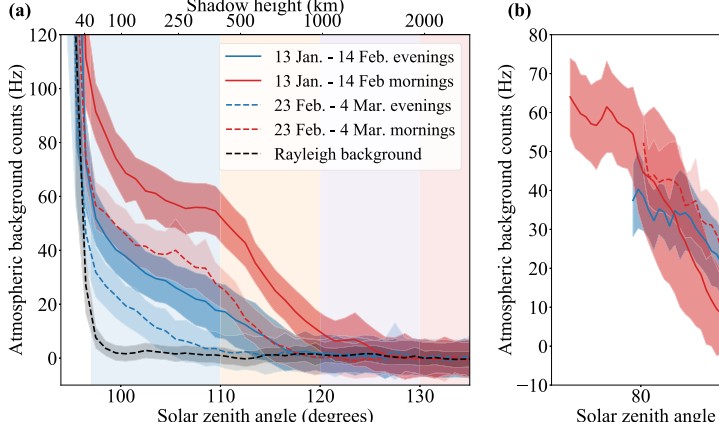

**Fig. 3 | Passive He(2³S) measurement. a** Atmospheric background counts as a function of solar zenith angle (SZA, bottom axis) or shadow height, i.e., the lowest height of the atmosphere illuminated by sunlight (top axis). Baseline background counts of 75–100 Hz (calculated from data with SZA > 125°) are subtracted independently from each night of measurements. Excess background is observed whenever the helium layer (below 1000 km) is illuminated. A strong asymmetry between morning (blue lines) and evening (red lines) is apparent, as well as a significant decrease between the first and second periods of measurements (solid vs. dashed lines). The data are averaged over 1° bins in SZA; the shaded regions about each line indicate the standard deviation within each bin. The dashed black line shows the contribution due to Rayleigh scattering in the lower atmosphere, as measured at a wavelength 70 pm off the He(2³S) line; this accounts for the rapid increase at SZA < 97°. The shaded bands in SZA indicate the binning intervals used in the active measurement (see Fig. 2). **b** Atmospheric background counts as a function of SZA at the geomagnetic conjugate point to the measurement location.

peak occurs at or above 500 km altitude; He(2³S) density decreases with increasing SZA; and He(2³S) density decreases as the season transitions from winter to spring, with a larger decrease at larger SZA. The evening density profiles show only a weak detection of He(2³S).

The passive measurements shown in Fig. 3a provide independent confirmation that He(2³S) was present in significant quantities during these measurements. Furthermore, they demonstrate several common characteristics: the signal is consistently higher in the morning as compared to the evening (red vs. blue lines), and the signal exhibits a significant decrease between the first and second periods of measurements (solid vs. dashed lines).

As has been highlighted in conjunction with OI excitation emission at 844.6 nm[25] (for which the impact of photoelectrons is also a major source), both the morning/evening asymmetry and the seasonal dependence of the He(2³S) brightness can be attributed to the influence of photoelectrons from the geomagnetic conjugate point, with the primary requirement that the conjugate point is illuminated while the local thermosphere is still in shadow. This condition is met in winter in both the morning and the evening, but due to the eastward

position of the conjugate point relative to Oberpfaffenhofen (35.2°S, 22.3°E according to the International Geomagnetic Reference Field model) it persists longer in the morning than in the evening. As the season advances from winter into spring, this condition is met progressively less, but the asymmetry remains. Figure 3b shows the passive He(2³S) measurement as a function of SZA at the conjugate point, demonstrating the dependence of He(2³S) brightness on the illumination of the conjugate point and reinforcing the conclusion that the production of He(2³S) is dominated by photoelectron impact from the conjugate point.

The measurements presented here establish He(2³S) as a viable target for resonance lidar measurements. The He(2³S) density profiles remove the largest experimental uncertainty, namely the lack of empirical validation of He(2³S) density predictions. Now that the approach has been validated, straightforward improvements to laser power and instrument efficiency will enable the implementation of well-established resonance lidar techniques to retrieve wind and temperature profiles by tuning the laser to multiple frequencies and measuring the resulting backscatter[20,21,26–28]. As shown in Fig. 1, the

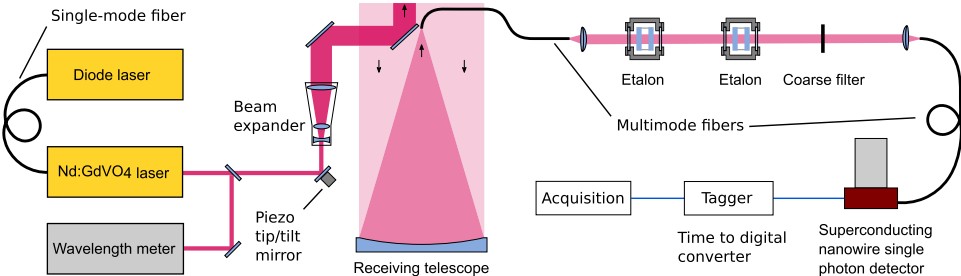

**Fig. 4 | Simplified schematic of the helium lidar.** The Nd:GdVO$_4$ laser, seeded by the continuous wave diode laser, emits narrow-band laser pulses at 1083 nm wavelength, which are released into the atmosphere. Backscattered light collected by the receiving telescope is spectrally filtered to suppress background radiation and detected using a superconducting nanowire single-photon detector.

spectral shape of the He(2$^3$S) scattering cross-section is sensitive to temperature; similarly, wind along the line-of-sight of the laser introduces an overall shift to the spectrum. Such multi-frequency measurements can therefore retrieve neutral wind, temperature, and density profiles simultaneously. For example, adding amplifier stages to increase the laser output power to 10 W while maintaining a bandwidth < 2 pm would yield estimated constraints of ± 40 m/s on line-of-sight winds, ±30 K on temperatures, and ±2.5% on He(2$^3$S) densities, for measurements with 1 h integration over 100 km vertical bins[28]. The ability to measure wind profiles will be particularly groundbreaking: wind measurements in the thermosphere to date are sparsely distributed in space and time, and no instrument thus far is capable of measuring neutral winds in the upper thermosphere at multiple altitudes. This observational gap has been specifically highlighted as a major obstacle to further improvements to space weather forecasts[3]. The helium lidar, therefore, represents a unique measurement capability that will provide invaluable insight and empirical validation to existing space weather models, increasing their accuracy and reliability. This, in turn, will lead to greater robustness and security for ground- and space-based systems on which society critically depends.

## Methods
### Instrument setup
Our setup (see Fig. 4) includes two important aspects which were key to the successful detection of He(2$^3$S): a powerful pulsed laser operating at the helium wavelength 1083 nm and the use of a superconducting nanowire single-photon detector[29] (SNSPD) in the receiver. The laser is a diode-pumped and Q-switched Nd:GdVO$_4$ laser developed at the German Aerospace Center. In order to achieve spectral narrow-band operation, the laser is seeded with a continuous wave diode laser, and the wavelength is monitored with a wavelength meter. To improve the coupling efficiency of the laser/telescope system, a beam expander is used to reduce the beam divergence before the beam is transmitted to the sky in a near-coaxial configuration with the receiving telescope. Overlap between the laser beam and the receiving telescope field of view is maintained using the conical scan method[30]. The light collected by the telescope is spectrally filtered by two narrow-band etalons in combination with a coarse interference filter. A narrow bandpass is necessary because, as shown above, the fluorescence of thermospheric He(2$^3$S) is very weak and can be easily washed out by the sky background if a broad-band receiver is used. Due to the telescope defocusing at low altitudes and relatively low laser output power, no mechanical chopper is required. The custom-made SNSPD detects the filtered light, and corresponding electrical pulses are time-stamped with nanosecond resolution. The timestamps are then joined with the firing time of the laser, and the timing information is stored for later reconstruction of lidar backscatter profiles. While SNSPDs are widely used in e.g., quantum communication, the application of this technology in lidar is relatively new[31–33]. This is despite the fact that SNSPDs are particularly suited to lidar due to high detection efficiency

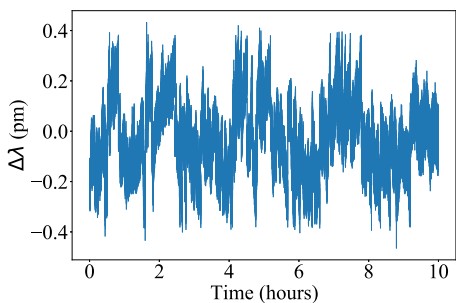

**Fig. 5 | Laser wavelength stability.** Laser output wavelength fluctuations about 1083.3300 nm (vacuum) as a function of time, over a single night's measurement.

(extending into the infrared regime), low noise, and high time resolution.

The laser is an in-house developed system that produces 200 ns laser pulses with 2.5 mJ pulse energy at 200 Hz pulse repetition frequency. It is injection-seeded using a distributed feedback laser diode (eagleyard Photonics). The wavelength is measured with a wavelength meter (TOPTICA WS6-200) and is locked to 1083.3300 nm (vacuum) by means of a feedback control loop. The RMS fluctuations of the wavelength are 0.1 pm (see Fig. 5), and the wavemeter has an absolute accuracy of 0.8 pm. The laser bandwidth was estimated to <2 pm using a 0.5 cm$^{-1}$ free spectral range etalon (LightMachinery Inc.). The photon detection system is a SNSPD system (Quantum Opus) with a custom 16 element chip, of which 12 elements are operable. The SNSPD is coupled to a 62.5 μm core diameter multimode fiber with an NA of 0.37 (OPTRAN WFGE62.5/87.5/110/125p37). The SNSPD is operated at a reduced bias current to limit dark counts; given the bias currents used here, its detection efficiency is ~30% at a wavelength of 1083 nm, with 70 Hz dark counts. The optical filters (see Fig. 4) comprise a 0.7 nm wide multi-cavity coarse filter (Alluxa) with an optical depth of 5, and two etalons with free spectral ranges at this wavelength of 0.618 and 1.00 nm, respectively, and each with a finesse of >22 and clear aperture of 30 mm (LightMachinery Inc.). Both etalons are air-gapped and tuned independently by changing the air pressure inside the etalon housing to align the transmission maxima with the He fluorescence wavelength. The temperature stability of the lab is ± 1°C, and the etalons each have a temperature sensitivity of 0.5 pm/°C, so temperature drift has a minimal effect on the transmission function of the receiver optics. The bandwidth (FWHM) of the combined filter assembly was measured to be 29.7 pm. The filters are coupled to the 762 mm diameter receiving telescope using a 105 μm core diameter multimode fiber with an NA of 0.22 (Thorlabs FG105LCA), resulting in a 61 μrad field of view.

### Data reduction
The raw data recorded by the helium lidar system consists of time-stamps, $t_i$, corresponding to arrival times of photons at the SNSPD,

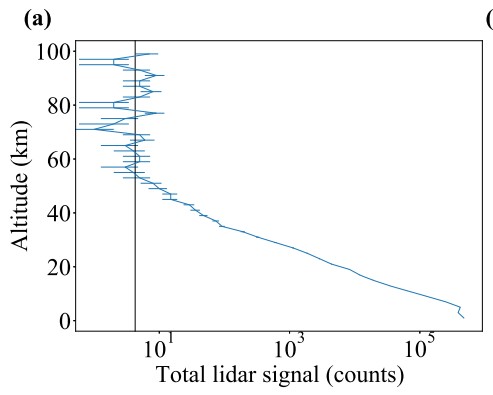

**(a)**

**(b)**

**Fig. 6 | Example raw lidar return plots. a** Example raw return count profile for a single 10 s interval with 2 km vertical resolution, showing the Rayleigh return to >50 km. The vertical black line shows the background estimate for these data. **b** Example raw return count profile for morning data with solar zenith angle between 97 and 110° from measurements taken between January 13 and February 14, 2022. This return count profile corresponds to the He(2³S) density profile plotted in blue in Fig. 2b and comprises 5.7 h of data. The vertical black line shows the background estimate for these data. The error bars represent 1σ uncertainties arising from photon noise.

together with the firing time, $t_0$, of the corresponding laser pulse. The scattering distance (or in our case, altitude), $z_i$, for a given photon is related to the time-of-flight by:

$$z_i = \frac{c(t_i - t_0)}{2} \qquad (1)$$

where $c$ is the speed of light.

The first step in our data reduction is to convert the timestamps into scattering altitudes and separate them into 2 km vertical bins to derive the number of detected photons as a function of altitude. To further reduce the data volume, we integrate these vertical profiles over 10 s intervals and record the number of pulses within each interval. An example of such a profile is shown in Fig. 6a. Note that a non-negligible fraction of the detected photons is due to sky brightness or detector noise, rather than laser backscatter, resulting in an altitude-independent 'background'. We estimate this background by averaging the return signal corresponding to altitudes between 100 and 250 km, where Rayleigh scattering is negligible, and no He(2³S) is expected to be present, due to efficient quenching by Penning ionization. We normalize for the varying pulse count within each interval and scale by $\frac{c}{2\tau\Delta z f}$ to convert to frequency, where $\tau = 10$ s is the integration time, $\Delta z = 2$ km is the altitude range of a single bin, and $f = 200$ Hz is the pulse repetition rate of the laser.

To remove spurious data arising from transient sources (e.g., lights in the laboratory) or rapidly changing receiver optics transmission during etalon tuning, we perform two cuts: first, we eliminate all profiles with a background higher than twice the median, calculated over a 10 min moving window. Next, we compare the average signal coming from above 300 km to the average background and eliminate profiles differing by more than three standard deviations from the median of this metric, again calculated over a 10-minute moving window.

Tropospheric clouds act as a contaminant, both by contributing excess background and by attenuating any signal, passive or active, originating above the cloud layer. To identify periods with significant cloud cover, we use as a metric the background-subtracted Rayleigh return signal from 30 to 40 km. We set a threshold of 40% of the maximum value of this metric and eliminate all profiles below this threshold. This also eliminates profiles with diminished signal due to all other causes (icing on mirror surfaces, misalignment between laser beam and telescope field-of-view, or reduced laser output power), which would otherwise reduce our detection significance.

**Metastable helium density profiles.** We retrieve He(2³S) density profiles from the data through the lidar equation[34]:

$$N(z, \tau, \Delta z) = \eta \left( \frac{P_L \tau}{E_\gamma} \right)(n_i \sigma_i \Delta z)\left( \frac{A}{4\pi z^2} \right) + \frac{2N_B \tau \Delta z f}{c} \qquad (2)$$

where

$N(z, \tau, \Delta z)$ is the number of detected photons corresponding to the altitude bin from $z - \frac{\Delta z}{2}$ to $z + \frac{\Delta z}{2}$ measured over time interval $\tau$;
$\eta$ is the end-to-end coupling efficiency of the lidar system;
$P_L$ is the power output of the laser;
$E_\gamma$ is the energy per photon;
$n_i$ is the number density of the scattering constituent;
$\sigma_i$ is the scattering cross-section of the scattering constituent;
$A$ is the receiving area of the telescope; and
$N_B$ is the range-independent background rate.

Note that $\eta$ incorporates a wide range of effects: imperfect overlap between the laser beam and the telescope field-of-view, atmospheric absorption, transmission losses at fiber surfaces or in the receiver optics, and imperfect detector efficiency, among others. Rather than characterizing each of these effects independently, we use the Rayleigh return from the lower atmosphere to calibrate the end-to-end efficiency of the system. We therefore obtain

$$n_{He}(z) = \frac{N_{He}(z) - \frac{N_B \tau \Delta z_{He} f}{c}}{N_R - \frac{N_B \tau \Delta z_R f}{c}} \frac{n_R \sigma_R \Delta z_R}{\sigma_{He} \Delta z_{He}} \frac{z^2}{z_R^2} \qquad (3)$$

where subscripts of 'He' refer to metastable helium, subscripts of 'R' refer to Rayleigh scattering, summed where relevant over the altitude range 36–50 km (with $\Delta z_R = 14$ km). We use the MSIS 2.0 model[35] to estimate the number density of the lower atmosphere.

This calibration process introduces several further potential biases into our He(2³S) density estimates. Firstly, the use of the Rayleigh signal to calibrate the return from the thermosphere assumes an altitude-independent coupling efficiency. However, focusing the telescope at 500 km results in decreased coupling efficiency in the lower atmosphere, and vice-versa. Measurements were made at a range of focus positions, and the focus drifted with temperature over the course of each night; rather than correct for this effect, we included it in our error estimates. This effect has a magnitude of up to 27%. Secondly, we assume an effective He(2³S) scattering cross-section of $2.78 \times 10^{-16}$ m², corresponding to a temperature of 1000 K and no vertical winds. Assuming a possible temperature range of 700–1500 K and vertical winds of up to 50 m/s, this introduces a further 11% uncertainty. Finally, variations in the laser wavelength and in the

number density of the lower atmosphere each contribute up to 2% uncertainty.

The 200 Hz repetition rate of the laser results in a 750 km separation between consecutive pulses. This has two consequences: (1) our background estimate from altitudes between 100 and 250 km will include a contribution from He(2$^3$S) backscatter from altitudes between 850 and 1000 km; and (2) our measured He(2$^3$S) backscatter at a given altitude will contain contributions from backscatter from altitudes 750 km higher. Given the expected decrease in He(2$^3$S) density at higher altitudes and the $z^{-2}$ dependence of the return signal on viewing distance, these contributions are expected to be small (<10%). Furthermore, the two effects counteract one another. In the present study, we account for these effects by including a 5% relative uncertainty in our calculated He(2$^3$S) densities; in future iterations, we will reduce the repetition rate of the laser to increase the separation between consecutive pulses and prevent pulse overlap.

The various errors associated with the calibration process and with the pulse overlap are independent and are therefore added in quadrature to give a 30% relative uncertainty. Our final error estimate is the quadrature addition of this relative error with the statistical uncertainties arising from photon noise.

We separate the filtered data into evening and morning measurements (i.e., before and after the sun reaches its minimum SZA) and by date (January 13–February 14, 2022 or February 23–March 4, 2022). Within each subset, we bin the data by SZA: 97–110°, 110–120°, 120–130°, and >130°, then calculate the number density of He(2$^3$S) as a function of altitude over 100 km altitude bins, according to the above formula. For reference, Fig. 6b shows the raw return count profile for data with SZA between 97 and 110° during the January 13–February 14, 2022 morning measurements (corresponding to the blue line in Fig. 2b), comprising 5.7 h of total measurements.

**Passive measurement.** For each measurement, we subtract a baseline level, calculated from the average background across all profiles with SZA greater than 125°. Note that this baseline was not constant across all measurements; after our second night of measurements, we improved the baffling of our receiver optics, thereby reducing the stray light contribution by 25%. By comparing the baseline with improved baffling to dark noise measurements of our detector, we can place an upper limit on contamination of our signal by OH airglow of <5 Hz. We bin the data in 1° intervals in SZA, and calculate the mean and standard deviation within each bin. The result is shown in Fig. 3a. To measure the background contribution from Rayleigh scattering in the lower atmosphere (black dashed line in Fig. 3a), the receiver optics were tuned 70 pm off the He(2$^3$S) line towards longer wavelengths. Assuming a He(2$^3$S) temperature of 1000 K and no vertical wind, this corresponds to an attenuation of the He(2$^3$S) signal by ~200x−thus, the resulting measurement effectively isolates the contribution to the background from Rayleigh scattering. The laser was not operated during this measurement, so no cut based on the Rayleigh return signal is applied. Fig. 3b was generated in an identical way, except that (1) the data are binned by SZA at the geomagnetic conjugate point rather than the local SZA, and (2) data with local SZA < 100° are excluded, to prevent contamination by Rayleigh scattering in the lower atmosphere.

## Data availability
Source data are provided in the Supplementary Data file. The data used in this study are available in the figshare database under accession code b68943b88521b1ce696d: https://figshare.com/s/b68943b88521b1ce696d Source data are provided with this paper.

## Code availability
The codes used to calculate He(2$^3$S) density and lidar return profiles and the passive He(2$^3$S) signal are available for download at https://github.com/geach89/HELIX and can be accessed and cited at: https://doi.org/10.5281/zenodo.7085329.

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

## Acknowledgements

This work was supported by internal funding from the German Aerospace Center.

## Author contributions
B.K. and M.R. conceived of the project. B.K. led the project, designed the lidar system, and wrote the control and data acquisition software. H.C.B. designed and built the laser. H.C.B. and A.M. designed and built the beam expander. A.M. designed the receiving telescope. B.K., C.G., and A.M. built the lidar and performed measurements. C.G. performed the data analysis. B.K. and C.G. wrote the manuscript with input from all co-authors. M.R. acquired the funding for the lidar project.

## Funding

## Competing interests
The authors declare no competing interests.
