## [Peer Review File · Nature Communications]

Measurements of metastable helium in Earth's atmosphere by resonance lidarREVIEWER COMMENTS

Reviewer #1 (Remarks to the Author):

This article describes a new lidar system that can detect the metastable helium in the atmosphere in the height of 200km-750km, which is claimed to be the first helium lidar until now. This method of active detection of atmospheric metastable helium using a pulsed 1083nm narrow linewidth laser was firstly proposed more than 20 years ago (Gerrard et. al, 1997). In Gerrard's work, they give an estimate example of helium lidar system, which has a laser power of 12W, detector's efficiency of 3%, telescope aperture of 0.66m², height resolution of 1km and an accumulation time of 10min. This system (if been realized) would have a SNR of about 2 when SZA is 104 in height of 400. In this reviewed paper, they realized a system, which has a laser power of 0.5W, detector's efficiency of 30%, telescope aperture of 0.45m², height resolution of 100km and an accumulation time of 240min (40mins/day*6days) and finally achieved a SNR of lower than 10 in the best condition (in the morning). Although they use a new powerful SNSPD (efficiency of 30%), but the laser power is too low (0.5W). Generally, this system's hardware is slightly weaker than the simulated system in Gerrard's work. And the signal is predictably weak (240mins of accumulating of 6days's data achieve an SNR lower than 10). The author claimed in the abstract that "These measurements open the door for the development of more sophisticated lidars: by applying well-established spectroscopic lidar techniques, one can measure the Doppler shift and broadening of the He(23S) line, thereby retrieving profiles of neutral 16 wind speed and temperature, opening a new window for studying space weather phenomena", however, I think in this kind of data quality (or SNR), this system's parameters are far away from measuring wind and temperature with an acceptable accuracy.

Besides the upper general comments on this paper, I would like to show some specific questions I have after reading this paper:

1>Line 84: the author claims this is the first time an SNSPD has been coupled to a large aperture lidar system. Although I am not 100% sure about whether there is this kind work realized by others. I think this point is not a worthy novel point to be claimed here. Because a aperture of 0.45m² can hardly be defined as an large aperture. In the meantime, SNSPD is widely used in quantum communication and other fields, it's not a novelty by simple use it into a lidar system.

2>Line 224: The in-house made 1083nm Nd:GdVO₄ laser is a new innovative technique. I think the author should show more testing results of this laser (in this paper they only said it's locked with an accuracy of 0.5pm), especially the frequency stability of this pulsed laser. That's because the long-term stability of the laser frequency can directly affect the helium measurements as the backscatter ratio is changing in different frequency. This work accumulates photon counts crossing 5-6 days, which means that the performance of the laser during these days should be carefully tested.

3>Line 228: The SNSPD from Quantum Opus has a custom chip. The author didn't mention how it's been customized. I tried to have a look at the raw data in the link (<https://figshare.com/s/b68943b88521b1ce696d>) provided by the author, and I find the raw data has 12 channels. The author didn't explain this in the paper. To my knowledge, usually a multi-element SNSPD has a $2*2=4$ elements or $4*4=16$ elements chip. If the 4 of the 16 elements is not active, this kind of condition will cause inhomogeneous energy distribution of the signal on the detectors, which should be carefully considered. Besides, I also have a question about the ultra-low dark counts (70Hz) of the multimode-fiber-coupled SNSPD. To our knowledge, multimode fiber with a large core diameter will bring large dark counts to the detector by emitting black body radiation, and the dark counts could probably be raised up to 1k per elements. Unless there is extra long-wavelength-filter mounted inside the cooling chamber of the SNSPD to block the black body radiation. This dark count issue should be explained in details.

4>Figure 4: Baseline background counts ($SZA > 125$) are subtracted. I don't think this is a necessary process. Considering the weak signal of metastable helium, the background noise is an important issue that should be shown and measured without too much nonobjective processing procedure. At least, the subtracted baseline background counts should be shown separately.

5>Figure 5: The author only shows the 6-days total accumulated photon counts. I think this is not very clear to help the reader to know the actual raw signal of this lidar. Single day's raw signal should be shown in different SZA. Besides, I think the raw photon counts figures should be in the main body of the paper, instead of in the supplement materials.

6>Line 231: about the dual-FPI filters, as we know, the FPI is very sensitive to the temperature. The FPI system frequency should be tuned and stabilized in a high accuracy. Is this system's temperature controlled? What's the temperature stability of this system? Besides, the author should mention how large the FPI's aperture is.

7>the observing position (Longitude and Latitude) of the system is not mentioned in this paper.

8>The laser pulse repetition rate is 200Hz, which means that when we receive the 750km signal of the Nth pulse, the next pulse (N+1th) will be emitted into to the sky. Furtherly, the backscatter of N+1th pulse in height 300km will be contaminated by the signal of the former pulse (Nth) at height of 1050km. According to the metastable helium model, there are possibly signals of metastable helium in around 1000km. This problem can't be solved unless reduce the laser repetition rate.

Reviewer #2 (Remarks to the Author):

Key Results

This paper presents the first height-resolved measurements of the metastable helium (He) density distribution in the Earth's upper thermosphere and furthers the potential for lidar remote sensing of atmospheric temperature and wind profiles near 500 km altitude. This new observational capability would fill a long-standing measurement gap, since no other optical sensing technique is sensitive to atmospheric state conditions in this region, which spans low earth orbit and is particularly responsive to disturbances associated with space weather.

Validity

The validity of the lidar remote sensing technique described in this work as a means to measure metastable He density is based on the detection of backscatter signal above background and on the strong agreement between the derived density profiles and previously published passive emission observations and photochemical models.

Significance

Knowledge of upper thermospheric composition, temperature, and wind fields is notoriously uncertain due to the nearly complete historical lack of reliable observations. Earth's metastable He population, which forms a tenuous but broad layer near 500 km, has long been an attractive candidate target for optical remote sensing of this key atmospheric region, and this paper presents the first direct observations of the metastable He density distribution as a function of altitude using active lidar techniques. The measurements reported in this work provide empirical constraints on the target metastable He population that are needed to guide the development of more sophisticated spectroscopic lidar systems. Such platforms would enable long-sought probing of crucial dynamical state parameters in support of advancing understanding of the Earth's response to space weather perturbations, which pose hazards to both ground and space based technologies.

Data and methodology

My expertise, which is primarily in the field of aeronomy and data science, does not allow me to evaluate the quality of the lidar measurements in terms of the experimental configuration or instrument calibration, and my comments here are restricted to the density retrieval approach. The methodology to derive metastable He number densities from the measured backscatter signal is appropriate overall. However, the stated use of the "MSIS-E-90 model" (line 285) to estimate number densities in the lower atmosphere is confusing, particularly given the lack of citation. This name appears to be a portmanteau of the decades-old "MSIS-90" model and the newer "NRLMSISE-00" model, though neither is considered state-of-the-art since the release of the MSIS 2.0 model last year. The MSIS-90 model is sufficiently outdated relative to NRLMSISE-00 and MSIS 2.0 that its use in equation (3) may systematically bias the

derived densities. In addition, the paper appropriately reports error bars on the derived densities in Figures 3 and 4, which account for "statistical uncertainty plus 30% relative uncertainty due to uncertainty in the calibration process" (lines 291-292). Presumably, "statistical uncertainty" refers to the Poisson-distributed shot noise on the detected photons, which would be appropriate, but without further details regarding the calibration process, the magnitude of "30%" is not justified in the manuscript.

Analytical approach

The authors provide a thorough assessment of the derived metastable He density distributions in terms of past observations and model analysis. They compellingly demonstrate that their lidar-derived metastable He profiles not only exhibit densities and peak altitudes that agree with photochemical model predictions, but also that their observed dependence on solar illumination (SZA) and season are consistent with expectations that the metastable He population is produced primarily through impact on ground-state He atoms by photoelectrons created through solar photoionization. Although the authors report the presence of a morning/evening asymmetry in both their observations and historical passive 1083nm emission radiance data, they note that "the reason for it remains unclear" (line 116). I disagree with this claim: both the seasonal dependence and local time asymmetry is likely associated with the geomagnetic field geometry in the conjugate hemisphere, whose solar illumination governs the supply of photoelectrons that excite the emission after sunset (or before sunrise) in the local hemisphere. This asymmetry was described in detail in Waldrop, Kerr, and Richards [JGR, 2008] based on mid-latitude observations of OI 844.6 nm emission, which is also excited primarily via photoelectron impact, and the morphological features of the metastable He density decay with increasing SZA shown in Figure 4 bear a striking resemblance to the twilight decay in O 844.6 nm emission radiance reported by those authors. However, the authors of this work did not report the location of the lidar experiment, and if the data was not acquired on a closed, mid-latitude field line, this explanation would be invalid.

Suggested improvements

Because the metastable He population is produced primarily by photoelectron impact, the authors should report the location of the lidar observations as well as the associated viewing geometry (e.g., zenith-oriented) in order to enable a more detailed evaluation of the SZA dependence and local time asymmetry reported in terms of geomagnetic field geometry. Also, if the metastable He densities were derived using the MSIS-90 model, they should be re-derived using the newest MSIS model version (2.0) in order to avoid the possibility of introducing bias due to outdated background atmospheric specification.

Clarity and context

This paper is well written and provides an excellent historical context and motivation for the work reported.

References

The references are complete and appropriate.

Reviewer #3 (Remarks to the Author):

Review of "Measurements of metastable helium in Earth's atmosphere by resonance lidar" by Kaifler et al.

The paper presents the first direct measurements of the metastable Helium profile in the earth's upper atmosphere. The results are significant in understanding the Earth's atmosphere, as well as representing a significant technical accomplishment on the part of the research team. The study of metastable Helium has recently been invigorated by observations of Helium in the tails of exoplanets (e.g., Deming, Nature, 2018).

The results are noteworthy, as they provide measurements of the metastable He profile in both morning and evening, on nights over a seven-week period. The lidar employs a novel laser and detector that will be of interest to the remote sensing community. The work will be of significance to the space physics community. Observations of the metastable Helium profile address atmospheric evolution and escape, large-scale transport in the upper atmosphere, the photoelectron flux, satellite drag. The presentation of the results is complete. The experimental method and interpretation of the observations is well explained. There are no flaws in the data analysis, interpretation or conclusions. The methodology is sound. The quality of the work high and meets the standards of the resonance lidar and upper atmosphere research communities.

I have several recommendations for revisions before recommending the paper for publication.

I have a request for Figure 5. The authors should show the entire raw lidar signal at 2 km resolution to allow readers to confirm the Rayleigh, resonance, and background components of the signal and independently confirm the estimates of the metastable Helium concentrations. The authors should also cite the value of the resonance scattering cross-section (peak cross-section, and effective cross-section).

The SZA bands of $\sim 10^\circ$ represent 1 hour intervals. Thus the authors have determined the metastable densities at intervals of 1 hour at a resolution of 100 km and errors of 30%. While the potential for

temperature and wind measurements is clear, I recommend that they provide a quantitative statement for the straight-forward improvements that would yield invaluable wind and temperature measurements.

The authors should cite the work of Raizada et al. (Geophysical Research Letters, 2020) showing detection of Calcium Ions by resonance lidar upto 180 km.

In the simplified schematic in Figure 2, there is no mention of a mechanical optical chopper in the receiver system. Is there a chopper in the receiver, or is the backscattered signal detected from the ground to 700 km directly? Please state this explicitly in the article.

There is a typo on line 45 "fore example into the the" should read "for example into the".

Reviewer #4 (Remarks to the Author):

The development of Helium (He) resonance lidar system is a step forward and the authors deserve an applause for attempting such challenging measurements. The use of new detector based on superconducting technology is very innovative. This manuscript is worthy of publication after some corrections and explanations of some of the methods implemented in the study are made clear.

My main suggestions are listed below:

- Abstract should mention the coordinates of the location where the observations took place. I could not find this information and had to guess based on the authors place of work. But that may not be correct, and the readers should find this information easily.

- Since most of the resonance lidars are currently for the mesospheric and upper thermospheric regions, and the results are presented in terms of wither UT or LT. However, since helium exists beyond the "normal" resonance lidar altitude ranges, it would be good to explain the use of new terminology of binning the data based on solar zenith angle. This can be confusing to many readers, who are not familiar with such processing. It has been used in passive resonance airglow measurements, where the shadow height is used to estimate the altitude profile of airglow emissions and infer concentrations of metals. The lack of geographical location, or local time information makes analysis difficult to understand. I recommend that authors add a few sentences describing the advantages of using solar

zenith angle as compared to LT or UT. For example, they can say that “even though at this location (give the coordinates), lower altitude are not illuminated by the sun, but higher altitudes are above the shadow height region. Thus, it is a better approach to bin the data in terms of SZA, which is related to the local time or shadow height through the relation (please give the expression or a reference)”.

- Line 33: “Penning of ionization...”, This sentence is not clear to me. What are authors trying to say?

- Line 45: Replace “fore” with ‘for’

- In the Data Reduction section, Lines 247 – 249 are confusing. The background noise is embedded in the backscattered signal for all the altitude ranges. Isn't this true? In order to estimate the noise, the authors have used an altitude region where no signal from either Helium or Rayleigh scattering is expected. This is correct. What authors are trying to say in the lines 247 – 249 is not clear. Please rewrite them.

The authors would like to thank the reviewers for their careful reading of the manuscript and their many insightful comments. We believe the current draft to be significantly improved as a result of their suggestions. Our responses to specific comments are below, in red.

REVIEWER COMMENTS

Reviewer #1 (Remarks to the Author):

This article describes a new lidar system that can detect the metastable helium in the atmosphere in the height of 200km-750km, which is claimed to be the first helium lidar until now. This method of active detection of atmospheric metastable helium using a pulsed 1083nm narrow linewidth laser was firstly proposed more than 20 years ago (Gerrard et. al, 1997). In Gerrard's work, they give an estimate example of helium lidar system, which has a laser power of 12W, detector's efficiency of 3%, telescope aperture of 0.66m², height resolution of 1km and an accumulation time of 10min. This system (if been realized) would have a SNR of about 2 when SZA is 104 in height of 400. In this reviewed paper, they realized a system, which has a laser power of 0.5W, detector's efficiency of 30%, telescope aperture of 0.45m², height resolution of 100km and an accumulation time of 240min (40mins/day*6days) and finally achieved a SNR of lower than 10 in the best condition (in the morning).

Although they use a new powerful SNSPD (efficiency of 30%), but the laser power is too low (0.5W). Generally, this system's hardware is slightly weaker than the simulated system in Gerrard's work. And the signal is predictably weak (240mins of accumulating of 6 days's data achieve an SNR lower than 10).

It is true that the SNR resulting from our measurement are lower than we had hoped, based on Gerrard et al.'s work. The principal reason for the discrepancy is that the background estimate presented there ignores two points that turned out to be important: they assume a detector with no dark counts and furthermore neglect to account for the passively stimulated He fluorescence that we show in Figure 4, thus assuming no background counts. The background of our instrument (dark counts plus passive background) results in a factor of ~100 more noise than what was assumed by Gerrard et al., with a corresponding decrease in our SNR and increase in the required integration time.

The author claimed in the abstract that "These measurements open the door for the development of more sophisticated lidars: by applying well-established spectroscopic lidar techniques, one can measure the Doppler shift and broadening of the He(23S) line, thereby retrieving profiles of neutral wind speed and temperature, opening a new window for studying space weather phenomena", however, I think in this kind of data quality (or SNR), this system's parameters are far away from measuring wind and temperature with an acceptable accuracy.

The reviewer is right in stating that the system as described in the paper is not capable of making meaningful wind or temperature measurements. But applying well-established techniques to increase the output power of the laser (namely, by adding amplifier stages) will enable such measurements to be made. A quantitative statement clarifying the necessary improvement and the resulting constraints has been added (see lines 145-147) in the revised manuscript).

Besides the upper general comments on this paper, I would like to show some specific questions I have after reading this paper:

1>Line 84: the author claims this is the first time an SNSPD has been coupled to a large aperture lidar system. Although I am not 100% sure about whether there is this kind work realized by others. I think this point is not a worthy novel point to be claimed here. Because a aperture of 0.45m² can hardly be defined as an large aperture. In the meantime, SNSPD is widely used in quantum communication and other fields, it's not a novelty by simple use it into a lidar system.

The reviewer is right that the phrase "large aperture" is not appropriate. By lidar standards, our aperture is indeed large, but by astronomy standards this is not the case. Nevertheless, given the complications that are present in coupling an SNSPD to a telescope of this size, we feel the novelty is worth highlighting. We have removed the phrase "large aperture" and replaced it with "aperture greater than 0.5 m in diameter".

2>Line 224: The in-house made 1083nm Nd:GdVO₄ laser is a new innovative technique. I think the author should show more testing results of this laser (in this paper they only said it's locked with an accuracy of 0.5pm), especially the frequency stability of this pulsed laser. That's because the long-term stability of the laser frequency can directly affect the helium measurements as the backscatter ratio is changing in different frequency. This work accumulates photon counts crossing 5-6 days, which means that the performance of the laser during these days should be carefully tested.

This is an important point that we did not highlight enough. The laser contains a feedback control loop to maintain a stable wavelength. The laser wavelength was continually monitored and remained within 0.5 pm of the desired value. A plot has been added to demonstrate the wavelength stability over a 10-hour period. See new Figure 5.

3>Line 228: The SNSPD from Quantum Opus has a custom chip. The author didn't mention how it's been customized. I tried to have a look at the raw data in the link (<https://figshare.com/s/b68943b88521b1ce696d>) provided by the author, and I find the raw data has 12 channels. The author didn't explain this in the paper. To my knowledge, usually a multi-element SNSPD has a 2*2=4 elements or 4*4=16 elements chip. If the 4 of the 16 elements is not active, this kind of condition will cause inhomogeneous energy distribution of the signal on the detectors, which should be carefully considered.

More information on the chip design has been added. The SNSPD is indeed a 16-element chip, of which 12 elements are active. The reviewer is right that the energy distribution across the chip is not uniform, but as we consider the total signal across all (active) channels, variations between channels do not constitute a source of error.

Besides, I also have a question about the ultra-low dark counts (70Hz) of the multimode-fiber-coupled SNSPD. To our knowledge, multimode fiber with a large core diameter will bring large dark counts to the detector by emitting black body radiation, and the dark counts could probably be raised up to 1k per elements. Unless there is extra long-wavelength-filter mounted inside the cooling chamber of the SNSPD to block the black body radiation. This dark count issue show be explained in details.

The SNSPD elements are operated at lower bias currents in order to reduce the dark counts. This also reduces the detection efficiency – this explains why the detection efficiency reported here is low relative to other SNSPDs. This detail has been added to the text.

Furthermore, the detector fiber has an anti-reflection coating for 1083 nm on the detector side - we have not seen the spectral performance of this coating, but it may act to further attenuate the longer wavelength black body radiation.

4>Figure 4: Baseline background counts (SZA>125) are subtracted. I don't think this is a necessary process. Considering the weak signal of metastable helium, the background noise is an important issue that should be shown and measured without too much nonobjective processing procedure. At least, the subtracted baseline background counts should be shown separately.

After our first two nights of measurements, we recognized that LEDs in our lab were contributing to our observed background. We improved the baffling, thereby reducing the baseline background by ~25%. To make the measurements from different nights more easily comparable, we subtract this baseline. We have added the approximate value of the subtracted baseline to the caption of the figure.

5>Figure 5: The author only shows the 6-days total accumulated photon counts. I think this is not very clear to help the reader to know the actual raw signal of this lidar. Single day's raw signal should be shown in different SZA. Besides, I think the raw photon counts figures should be in the main body of the paper, instead of in the supplement materials.

We believe the figure will be more useful if the helium peak is clearly visible. This is not the case in single day's profiles (hence our decision to bin over multiple days). We also believe that the placement of this plot in the supplemental materials is appropriate, since we use it to illustrate a step in our analysis pipeline (which is described in the supplemental materials). To give readers a sense of the magnitude of the return signal, the total duration of the plotted data is stated in the text; that information has now been repeated in the caption. We include an example of single day raw photon profiles, above, for reference.

6>Line 231: about the dual-FPI filters, as we know, the FPI is very sensitive to the temperature. The FPI system frequency should be tuned and stabled in a high accuracy. Is this system's temperature controlled? What's the temperature stability of this system? Besides, the author should mention how large the FPI's aperture is.

The etalons have a clear aperture of 30 mm and a temperature stability of 0.5 pm/degree C. Given the temperature stability of the lab (+/- 1 degree C), the temperature drift of the etalons is negligible compared to the FWHM of the receiver optics.

This information has been added to the text.

7>the observing position (Longitude and Latitude) of the system is not mentioned in this paper.

Thank you for highlighting this. This information has been added.

8>The laser pulse repetition rate is 200Hz, which means that when we receive the 750km signal of the Nth pulse, the next pulse (N+1th) will be emitted into to the sky. Furtherly, the backscatter of N+1th pulse in height 300km will be contaminated by the signal of the former pulse (Nth) at height of 1050km. According to the metastable helium model, there are possibly signals of metastable helium in around 1000km. This problem can't be solved unless reduce the laser repetition rate.

This is a good point, and one that should be highlighted in the text. The effect is likely to be small though, since two factors reduce the contributions from higher altitudes: 1) the metastable helium density is expected to decrease above its peak at ~500-600 km (see for example Fig 14. In Waldrop et al, 2005), and 2) the contribution to the backscatter signal decreases with the viewing distance squared. Thus, the contribution from higher altitudes to the 300-400 km bin is expected to be less than 10%; the contribution to higher bins will be even less.

The text has been modified to highlight this consideration; see lines 317-325.

Reviewer #2 (Remarks to the Author):

Key Results

This paper presents the first height-resolved measurements of the metastable helium (He) density distribution in the Earth's upper thermosphere and furthers the potential for lidar remote sensing of atmospheric temperature and wind profiles near 500 km altitude. This new observational capability would fill a long-standing measurement gap, since no other optical sensing technique is sensitive to atmospheric state conditions in this region, which spans low earth orbit and is particularly responsive to disturbances associated with space weather.

Validity

The validity of the lidar remote sensing technique described in this work as a means to measure metastable He density is based on the detection of backscatter signal above background and on the strong agreement between the derived density profiles and previously published passive emission observations and photochemical models.

Significance

Knowledge of upper thermospheric composition, temperature, and wind fields is notoriously uncertain due to the nearly complete historical lack of reliable observations. Earth's metastable He population, which forms a tenuous but broad layer near 500 km, has long been an attractive candidate target for optical remote sensing of this key atmospheric region, and this paper presents the first direct observations of the metastable He density distribution as a function of altitude using active lidar techniques. The measurements reported in this work provide empirical constraints on the target metastable He population that are needed to guide the development of more sophisticated spectroscopic lidar systems. Such platforms would enable long-sought probing of crucial dynamical state parameters in support of advancing understanding of the Earth's response to space weather perturbations, which pose hazards to both ground and space based technologies.

Data and methodology

My expertise, which is primarily in the field of aeronomy and data science, does not allow me to evaluate the quality of the lidar measurements in terms of the experimental configuration or instrument calibration, and my comments here are restricted to the density retrieval approach. The methodology to derive metastable He number densities from the measured backscatter signal is appropriate overall. However, the stated use of the "MSIS-E-90 model" (line 285) to estimate number densities in the lower atmosphere is confusing, particularly given the lack of citation. This name appears to be a portmanteau of the decades-old "MSIS-90" model and the newer "NRLMSISE-00" model, though neither is considered state-of-the-art since the release of the MSIS 2.0 model last year. The MSIS-90 model is sufficiently outdated relative to NRLMSISE-00 and MSIS 2.0 that its use in equation (3) may systematically bias the derived densities.

The reviewer is right that we had been using an outdated atmospheric model. We now use the MSIS 2.0 model, and have updated the figures and text to reflect this.

In addition, the paper appropriately reports error bars on the derived densities in Figures 3 and 4, which account for "statistical uncertainty plus 30% relative uncertainty due to uncertainty in the calibration process" (lines 291-292). Presumably, "statistical uncertainty" refers to the Poisson-distributed shot noise on the detected photons, which would be appropriate, but without further details regarding the calibration process, the magnitude of "30%" is not justified in the manuscript.

More detail has been added to describe the various contributions to the 30% calibration uncertainty (see lines 307-328).

Analytical approach

The authors provide a thorough assessment of the derived metastable He density distributions in terms of past observations and model analysis. They compellingly demonstrate that their lidar-derived metastable He profiles not only exhibit densities and peak altitudes that agree with photochemical model predictions, but also that their observed dependence on solar illumination (SZA) and season are consistent with expectations that the metastable He population is produced primarily through impact on ground-state He atoms by photoelectrons created through solar photoionization. Although the authors report the presence of a morning/evening asymmetry in both their observations and historical passive 1083nm emission radiance data, they note that "the reason for it remains unclear" (line 116). I disagree with this claim: both the seasonal dependence and local time asymmetry is likely associated with the geomagnetic field geometry in the conjugate hemisphere, whose solar illumination governs the supply of photoelectrons that excite the emission after sunset (or before sunrise) in the local hemisphere. This asymmetry was described in detail in Waldrop, Kerr, and Richards [JGR, 2008] based on mid-latitude observations of OI 844.6 nm emission, which is also excited primarily via photoelectron impact, and the morphological features of the metastable He density decay with increasing SZA shown in Figure 4 bear a striking resemblance to the twilight decay in O 844.6 nm emission radiance reported by those authors. However, the authors of this work did not report the location of the lidar experiment, and if the data was not acquired on a closed, mid-latitude field line, this explanation would be invalid.

We thank the reviewer for providing this reference. We agree that the description provided there explains the observed morning/evening asymmetry perfectly. We have re-written the discussion of Figure 4 and added a second subfigure to highlight this dependence on the state of illumination of the conjugate point. We have also added the location of the measurements presented here. See lines 126-136).

Suggested improvements

Because the metastable He population is produced primarily by photoelectron impact, the authors should report the location of the lidar observations as well as the associated viewing geometry (e.g., zenith-oriented) in order to enable a more detailed evaluation of the SZA dependence and local time asymmetry reported in terms of geomagnetic field geometry. Also, if the metastable He densities were derived using the MSIS-90 model, they should be re-derived using the newest MSIS model version (2.0) in order to avoid the possibility of introducing bias due to outdated background atmospheric specification.

These changes have been implemented (see detailed responses above).

Clarity and context

This paper is well written and provides an excellent historical context and motivation for the work reported.

References

The references are complete and appropriate.

Reviewer #3 (Remarks to the Author):

Review of "Measurements of metastable helium in Earth's atmosphere by resonance lidar" by Kaifler et al.

The paper presents the first direct measurements of the metastable Helium profile in the earth's upper atmosphere. The results are significant in understanding the Earth's atmosphere, as well as representing a significant technical accomplishment on the part of the research team. The study of metastable Helium has recently been invigorated by observations of Helium in the tails of exoplanets (e.g., Deming, Nature, 2018).

The results are noteworthy, as they provide measurements of the metastable He profile in both morning and evening, on nights over a seven-week period. The lidar employs a novel laser and detector that will be of interest to the remote sensing community. The work will be of significance to the space physics community. Observations of the metastable Helium profile address atmospheric evolution and escape, large-scale transport in the upper atmosphere, the photoelectron flux, satellite drag. The presentation of the results is complete. The experimental method and interpretation of the observations is well explained. There are no flaws in the data analysis, interpretation or conclusions. The methodology is sound. The quality of the work high and meets the standards of the resonance lidar and upper atmosphere research communities.

I have several recommendations for revisions before recommending the paper for publication.

I have a request for Figure 5. The authors should show the entire raw lidar signal at 2 km resolution to allow readers to confirm the Rayleigh, resonance, and background components of the signal and independently confirm the estimates of the metastable Helium concentrations.

Unfortunately, the resonance signal is not visible when the raw lidar signal is binned at 2 km resolution (hence our decision to bin over 100 km intervals): we include the requested plot here (above), showing the lidar return profile averaged over 5.7 hours at 2 km vertical resolution, over the entire range of return counts (left) and with the axis zoomed to show the 100 – 750 km signal in more detail. The vertical black line in both plots shows the background estimate for this interval.

We have also added a plot to the paper (right panel of what is now Figure 6) showing the lidar return at lower altitudes to demonstrate the Rayleigh signal, integrated over 10 seconds and plotted with 2 km resolution.

The authors should also cite the value of the resonance scattering cross-section (peak cross-section, and effective cross-section).

The cross-section is temperature dependent, and the temperature of the thermosphere can vary significantly. We now state the value for the cross-section that we assume (lines 312-313). Additionally, the temperature dependence of the cross-section is shown in the right panel of Figure 1.

The SZA bands of $\sim 10^\circ$ represent 1 hour intervals. Thus the authors have determined the metastable densities at intervals of 1 hour at a resolution of 100 km and errors of 30%. While the potential for temperature and wind measurements is clear, I recommend that they provide a quantitative statement for the straight-forward improvements that would yield invaluable wind and temperature measurements.

We have added text to describe what improvement is necessary and constraints on wind, temperature and density such an instrument would be capable of (see lines 145-147).

The authors should cite the work of Raizada et al. (Geophysical Research Letters, 2020) showing detection of Calcium Ions by resonance lidar upto 180 km.

We thank the reviewer for alerting us to the existence of this paper. The text has been updated with this reference.

In the simplified schematic in Figure 2, there is no mention of a mechanical optical chopper in the receiver system. Is there a chopper in the receiver, or is the backscattered signal detected from the ground to 700 km directly? Please state this explicitly in the article.

No mechanical chopper has been used – this is now stated explicitly in the text.

There is a typo on line 45 "fore example into the the" should read "for example into the".

This has been fixed.

Reviewer #4 (Remarks to the Author):

The development of Helium (He) resonance lidar system is a step forward and the authors deserve an applause for attempting such challenging measurements. The use of new detector based on superconducting technology is very innovative. This manuscript is worthy of publication after some corrections and explanations of some of the methods implemented in the study are made clear.

My main suggestions are listed below:

- Abstract should mention the coordinates of the location where the observations took place. I could not find this information and had to guess based on the authors place of work. But that may not be correct, and the readers should find this information easily.

Given the limited allowed length of the abstract, we have added this information to the main text of the manuscript.

- Since most of the resonance lidars are currently for the mesospheric and upper thermospheric regions, and the results are presented in terms of either UT or LT. However, since helium exists beyond the “normal” resonance lidar altitude ranges, it would be good to explain the use of new terminology of binning the data based on solar zenith angle. This can be confusing to many readers, who are not familiar with such processing. It has been used in passive resonance airglow measurements, where the shadow height is used to estimate the altitude profile of airglow emissions and infer concentrations of metals. The lack of geographical location, or local time information makes analysis difficult to understand. I recommend that authors add a few sentences describing the advantages of using solar zenith angle as compared to LT or UT. For example, they can say that “even though at this location (give the coordinates), lower altitudes are not illuminated by the sun, but higher altitudes are above the shadow height region. Thus, it is a better approach to bin the data in terms of SZA, which is related to the local time or shadow height through the relation (please give the expression or a reference)” .

We have added a justification for our use of SZA rather than local or universal time (see lines 100-102).

- Line 33: “Penning of ionization...” , This sentence is not clear to me. What are authors trying to say?

We are describing the vertical distribution of He(2^3S) and the processes that are responsible for its distribution. We have changed “lower atmosphere” to “below 250 km” and re-ordered the sentence slightly.

- Line 45: Replace “fore” with ‘for’

Done.

- In the Data Reduction section, Lines 247 - 249 are confusing. The background noise is embedded in the backscattered signal for all the altitude ranges. Isn't this true? In order to estimate the noise, the authors have used an altitude region where no signal from either Helium or Rayleigh scattering is expected. This is correct. What authors are trying to say in the lines 247 - 249 is not clear. Please rewrite them.

The text was superfluous and has been removed.

REVIEWERS' COMMENTS

Reviewer #2 (Remarks to the Author):

The authors have sufficiently addressed the questions and concerns raised in my initial review, and I believe the current manuscript is appropriate for publication without further revision.

Reviewer #3 (Remarks to the Author):

Review of "Measurements of metastable helium in Earth's atmosphere by resonance lidar" by Kaifler et al.

I thank the authors for their response to my review and revision on the manuscript.

1) Lidar signal

In response to my request to show the total lidar signal, the authors have included a new Figure 6, that shows the returns upto 100 km primarily due to Rayleigh scatter, and the returns from the Helium layer. One small point is that the figures might be better labelled on the x-axis as "Total Lidar Signal" in units of "Photon Counts".

2) Cross Section

The authors cite a cross section of $2.55 \times 10^{-16} \text{ m}^2$ (Line 313). I assume from Figure 1, this is the peak cross-section of the Helium at 1000 K, and not the effective cross section representing the convolution of the laser line shape and the Helium line shape. Can the authors clarify this? As regards Figure 1 the units of cross section on the y-axis appear to be " m^{-2} " not " m^2 ".

3) Winds and Temperatures

The authors have estimated the performance of a Lidar system with the addition of amplifier stages for measurement of temperature and wind (Lines 146-147). However, the authors cite a laser bandwidth of 2 pm. I assume this is the laser linewidth, corresponding to ~ 500 MHz. Will the addition of the amplifiers serve to broaden the line, and and is this sufficient for temperature and wind measurements. If

linewidth is an issue I request the authors add a brief statement about the target line width, not just the power scaling, to achieve reasonable wind and temperature measurements.

4) Receiver Architecture

The authors have clarified the receiver architecture, and added a sentence stating that there is no chopper in the system (Line 79).

5) Cancellation of Inter-Pulse Interference

The authors note that the effect of high altitude resonance signal from the previous laser pulse is to add to both the background signal and the resonance signal. The authors note that these contributions "act in opposite directions". Given that the background is calculated 100-250 km and thus includes a positive bias with contributions from 850-1000 km and the signal from 250-650 km thus includes a positive bias with contributions from 1000-1400 km, is the "opposite direction" based on the fact that the background and total signal are both increased, and thus when the background is subtracted, the bias is removed or at least reduced?

The authors would again like to thank the reviewers for their renewed efforts and further suggestions for improvements. Our responses to specific comments are below, in red.

REVIEWERS' COMMENTS

Reviewer #2 (Remarks to the Author):

The authors have sufficiently addressed the questions and concerns raised in my initial review, and I believe the current manuscript is appropriate for publication without further revision.

Reviewer #3 (Remarks to the Author):

Review of "Measurements of metastable helium in Earth's atmosphere by resonance lidar" by Kaifler et al.

I thank the authors for their response to my review and revision on the manuscript.

1) Lidar signal

In response to my request to show the total lidar signal, the authors have included a new Figure 6, that shows the returns upto 100 km primarily due to Rayleigh scatter, and the returns from the Helium layer. One small point is that the figures might be better labelled on the x-axis as "Total Lidar Signal" in units of "Photon Counts".

We agree with the suggestion and have implemented it.

2) Cross Section

The authors cite a cross section of $2.55 \times 10^{-16} \text{ m}^2$ (Line 313). I assume from Figure 1, this is the peak cross-section of the Helium at 1000 K, and not the effective cross section representing the convolution of the laser line shape and the Helium line shape. Can the authors clarify this? As regards Figure 1 the units of cross section on the y-axis appear to be " m^{-2} " not " m^2 ".

This comment has alerted us to an error in our calculation of the cross-section, which we have now corrected. The text now reads, "we assume an effective He(23S) scattering cross-section of $2.78 \times 10^{-16} \text{ m}^2$..." (line 228 in the revised manuscript).

Additionally, we have corrected the typo in the figure.

3) Winds and Temperatures

The authors have estimated the performance of a Lidar system with the addition of amplifier stages for measurement of temperature and wind (Lines 146-147). However, the authors cite a laser bandwidth of 2 pm. I assume this is the laser linewidth, corresponding to $\sim 500 \text{ MHz}$. Will the addition of the amplifiers serve to broaden the line, and and is this sufficient for temperature and wind measurements. If linewidth

is an issue I request the authors add a brief statement about the target line width, not just the power scaling, to achieve reasonable wind and temperature measurements.

We do not expect that the addition of pump chambers will increase the laser linewidth. Nevertheless, we have clarified the requirement on the target linewidth (line 126 in the revised manuscript).

“For example, adding amplifier stages to increase the laser output power to 10 W while maintaining a bandwidth <2 pm would yield estimated constraints of ± 40 m/s on line-of-sight winds, ± 30 K on temperatures, and $\pm 2.5\%$ on He(23S) densities, for measurements with 1 hour integration over 100 km vertical bins.”

4) Receiver Architecture

The authors have clarified the receiver architecture, and added a sentence stating that there is no chopper in the system (Line 79).

Correct.

5) Cancellation of Inter-Pulse Interference

The authors note that the effect of high altitude resonance signal from the previous laser pulse is to add to both the background signal and the resonance signal. The authors note that these contributions "act in opposite directions". Given that the background is calculated 100-250 km and thus includes a positive bias with contributions from 850-1000 km and the signal from 250-650 km thus includes a positive bias with contributions from 1000-1400 km, is the "opposite direction" based on the fact that the background and total signal are both increased, and thus when the background is subtracted, the bias is removed or at least reduced?

Yes, this is precisely what we mean. We have amended the text to make this clearer (line 237-238 in the revised manuscript).

“Furthermore, the two effects counteract one another.”